# Prevalence of and Factors Associated with Cardiometabolic Risks and Lung Function Impairment among Middle-Aged Women in Rural Taiwan

**DOI:** 10.3390/ijerph17218067

**Published:** 2020-11-02

**Authors:** Ming-Shyan Lin, Mei-Hua Yeh, Mei-Yen Chen

**Affiliations:** 1Department of Cardiology, Chang Gung Memorial Hospital, Yunlin 638, Taiwan; mingshyan@cgmh.org.tw; 2Department of Respiratory Therapy, Chang Gung Memorial Hospital, Yunlin 638, Taiwan; yeimeihua@gmail.com; 3Department of Nursing, Chang Gung University of Science and Technology, Chiayi 613, Taiwan; 4School of Nursing, Chang Gung University, Taoyuan 333, Taiwan; 5Department of Cardiology, Chang Gung Memorial Hospital, Yunlin 638, Taiwan

**Keywords:** middle age, women, lung function impairment, cardiometabolic risks, metabolic syndrome

## Abstract

Background: This study aims to explore the prevalence of and factors associated with cardiometabolic risks and lung function impairment among middle-aged women. Methods: A nurse-led community health development and cross-sectional study design was applied in Yunlin County, Taiwan. Lung function test was performed by a certified technician using a valid spirometer, during annual community health checkups conducted by a collaborating local hospital. Lung function impairment and cardiometabolic risks were measured and defined, based on the medical diagnosis and the national standard, by the hospital. Results: From a total of 439 middle-aged women, the prevalence of lung function impairment and metabolic syndrome were 26% and 47.2%, respectively. Many women adopted few health habits, e.g., only 30.5% engaged in regular exercise. A significant association between lung function impairment and four cardiometabolic risk factors (*p* < 0.05) was found. The multivariate logistic regression analysis showed that adopting few exercises (OR = 0.56, 95% CI 0.36–0.87) and lung function impairment (OR = 2.12, 95% CI 1.34–3.35) were independently associated with metabolic syndrome, after adjusting for confounding factors, such as age and education. Conclusions: The findings revealed that middle-aged women have a high prevalence of cardiometabolic risks and lung function impairment. Lung function impairment and physical inactivity were independently associated with an increased risk of having metabolic syndrome.

## 1. Introduction

Literature indicated that middle-age is used to describe the transition between early and late adulthood; most of the ages ranged from 40 to 64 years [1]. Some studies on middle age reported that many women were perceived younger than they actually were and wanting to be younger than their chronological age [1], and a youthful subjective age was found to be an indicator of successful aging and higher life satisfaction [2]. Owing to the increasing age and cumulative unhealthy habits, some chronic diseases soundlessly increased, such as diabetes and cardiometabolic risks [3]. Previous studies indicated the increasing prevalence of cardiometabolic diseases; for example, hypertension, diabetes, and metabolic syndrome were found in middle-aged adults [4,5,6]. Declining physical functions, poor fitness, and increased stress resulting from family and work obligations create important health challenges among middle-aged adults [7,8]. Additionally, midlife fitness could predict less burden of chronic disease in later life [7,9]. Therefore, early detection of cardiometabolic risk factors and maintaining good, healthy behaviors are particularly important for this age population. 

Cardiometabolic risk factors are a cluster of risk factors, including abdominal obesity, high fasting blood glucose (FBG), hypertension, high triglycerides, low high-density lipoprotein-cholesterol (HDL-C), and a lack of consumption of fruits (at least two portions) and vegetables (at least three portions), and inactive lifestyle that increase the risk of cardiovascular disease and diabetes [10,11,12]. In addition, these risk factors are associated with impairment of lung function [13,14]; lung function impairment leads to a high risk of low muscle mass in an older population [14,15]. However, few studies explored the lung function status in middle-aged women. Further, some studies indicated that inadequate fluid consumption was associated with rapid renal decline, cardiovascular disease mortality, and all-cause mortality in elderly women due to an increased level of blood viscosity [16]. Previous studies showed that providing lifestyle intervention could have a positive impact on healthy lifestyle behaviors, dietary intake, weight loss, blood pressure, triglyceride, total cholesterol, and HDL levels in middle-aged adults [6]. Hence, the early detection and the prevention of cardiometabolic risks, as well as maintaining optimal lung functioning were important tasks for the middle-aged population. These factors are best identified by primary care providers, as most patients present no symptoms at an early stage [17].

According to the official report in Taiwan [15], the recent average female’s life expectancy at birth was 84 years; however, the healthy life expectancy at birth was 74.7 years. That means women’s nearly 10 years were disabled and of less quality of life before death. There is a gap between life expectancy and healthy life expectancy. Healthy life expectancy is a useful indicator of a population’s overall health, reflecting the length of life as well as the quality of life; it also refers to an individual’s length of life lived without limitations in daily activities [18]. In an aging society, as greater age puts increased pressure on social systems, extending a healthy life expectancy and shortening life expectancy with disability are becoming global priorities [19]. Further, in middle age, higher cardiorespiratory fitness is strongly associated with lower health-care costs, at an average of 22 years later in life, independent of cardiovascular risk factors [20]. Many studies pointed out the importance of promoting healthy nutrition, diet, and physical activity for the prevention of cardiometabolic diseases in middle-aged adults [6,8,9,21]. Therefore, this study aimed to explore the prevalence of and factors associated with cardiometabolic risks and lung function impairment, among middle-aged women, especially in disadvantaged areas. 

## 2. Materials and Methods

### 2.1. Design, Sample, and Setting

This study was a series of reports from a nurse-led community health development program in adults living in rural areas of western coastal, Yunlin County, Taiwan. Data were drawn from a community-based cross-sectional study design and in collaboration with a local hospital, through an annual community health screening between March and December 2019. The inclusion criteria were (1) women aged 40–64 years, (2) able to communicate in Mandarin or Taiwanese, (3) fully independent in daily activities and able to walk to the community activity center, and (4) agreed to sign the informed consent form. The exclusion criteria were incomplete data and data without an informed consent form.

### 2.2. Procedure and Ethical Considerations

After approval from the Institutional Review Board (IRB NO: 201900222A3), informed consent was obtained from each participant by the research team. Six research assistants were trained for two hours by an investigator. They were all senior nursing students and were divided into three pairs to interview each pilot participant. A 90% correct rate of inter-rater reliability was confirmed. The purpose of the study and the data collection process was explained to all participants. For instance, 10 min for individual interviews of the structured questionnaire, blood sample collection after 8 h of overnight fasting to understand the cholesterol and blood glucose, and 5–10 min for the procedure of pulmonary function test to explore lung function impairment.

### 2.3. Measurements

Demographic characteristics: These included age, educational level (years), working status (have or have no job), living arrangement (live alone or with someone else), smoking habit (current or former smoker), and marital status.Health-related behaviors: Four adequate nutrition intake and engagement of regular exercise were measured and recommended by the literatures [4,17]: “Do you have at least three portions of vegetables per day? Do you have two portions of fruit per day? Do you have at least 1500 mL of water per day? Do you have engaged in regular exercise at least three times per week and more than 30 min each time?”Lung function status: This was diagnosed by a physician in the collaborated hospital. The automated flow-sensing spirometer was used and performed by a certified respiratory therapist (RT) in the community activity center. All participants were asked to stand and use a dry rolling-seal spirometer calibrated using one or three liters of precision syringe daily. According to the international standard [22,23], the normal lung function was defined as FEV1/FVC ≥ 70% and FVC ≥ 80%. Restrictive lung impairment was defined as an FVC < 80% of the predicted value and an FEV1/FVC > 70%. Obstructive lung impairment was defined as an FEV1/FVC ratio < 70% and FVC > 80% of the predicted value.Cardiometabolic risk factors and metabolic syndrome: Based on the national standard [4], five abnormal physiological biomarkers were measured: (1) Waist circumference > 80 cm; (b) systolic/diastolic blood pressure ˃ 130/85 mmHg; (c) HDL-C ˂ 50 mg/dL (1.29 mmol/L); (d) FBG level ˃ 100 mg/dL (5.6 mmol/L); and (e) triglyceride level ˃ 150 mg/dL (1.7 mmol/L). Three or more of these risk factors were defined as metabolic syndrome (MetS) [4].

### 2.4. Data Analysis

We compared the cardiometabolic risk factors between two groups of lung function status using the chi-square test for categorical variables or independent sample *t*-test for continuous variables. The assumption of normality of the continuous variables (age, education level, and mean number of cardiometabolic risks) was assessed using the Kolmogorov–Smirnov test and none of the tests were statistically significant. The binary logistic regression was used to explore the association between demographic characteristics, health-related behaviors, and lung function impairment, and the risk of metabolic syndrome. All tests were 2-tailed, and *p* < 0.05 was considered to be statistically significant. No adjustment of multiple testing (multiplicity) was made in this study. Data analyses were conducted using SPSS 25 (IBM SPSS Inc., Chicago, IL, USA).

## 3. Results

A total of 1054 adult women participated in this study. Table 1 shows that a total of 439 middle-aged women (40–64 years) with complete data were analyzed. The average age was 55.9 years (standard deviation [SD] = 7.1 years), and the mean education level was 8.7 years (SD = 4.5 years). Nearly half (49.9%) had a job; 6.4% lived alone. More than two-thirds (72.2%) reported having vegetable intake ≥ 3 portions per day; 64%, fruit intake equal to 2 portions per day; and 60.6%, water intake ≥ 1500 cc per day. However, only 30.5% engaged in regular exercise. The proportion of smoking was relatively few (3.6%). More than one-fourth of the participants had lung function impairment. More than half of the participants had cardiometabolic risk factors, including abnormal waist circumference (55.6%), hypertension (54.7%), and high FBG (67.4%). Nearly half of the participants (47.2%) had metabolic syndrome, and the average number of cardiometabolic risk factors was 2.5 (SD = 1.4).

Table 2 shows that the prevalence of lung impairment was significantly associated with hypertension (63.2% vs. 51.7%, *p* < 0.05), low HDL-C (45.6% vs. 32.3%, *p* < 0.05), high FBG (79.8% vs. 63.1%, *p* < 0.01), high triglyceride (45.6% vs. 28%, *p* < 0.01), and metabolic syndrome (62.3% vs. 41.8%, *p* < 0.001). 

Further, after adjustment of age and education, regular exercise (odds ratio [OR] = 0.56, 95% confidence interval [CI] 0.36–0.87) and lung function impairment (OR = 2.12, 95% CI 1.34–3.35) were significantly associated with metabolic syndrome (Table 3).

## 4. Discussion

The purpose of this study was to explore the prevalence of and factors associated with cardiometabolic risks and lung function impairment among middle-aged women. Three valuable findings were revealed in this study. First, a high prevalence of cardiometabolic risks, lung function impairment, and metabolic syndrome were found in middle-aged women in the disadvantaged areas. Second, lung function impairment was significantly associated with cardiometabolic risk factors. Finally, physical inactivity and lung function impairment were independently associated with metabolic syndrome. 

This study compared the general population of cardiometabolic risks with age–sex distribution from the fourth wave of Nutrition and Health Survey in Taiwan, between 2013 and 2016 [15], with a prevalence rate of high waist circumference (55.8%), high FBG (45.9%), low HDL (25.2%), high triglyceride (24.3%), hypertension (49.6%), and metabolic syndrome (36%) in 45–64 adult women. The present findings of this study indicated that rural middle-aged women had a higher prevalence of hypertension, low HDL-C, triglyceride, and metabolic syndrome than the general population in Taiwan. The prevalence of female smoking habits (3.6%) was similar to a previous investigation in the general Taiwanese population (4.3%) [24]. Further, in the present study, women with cardiometabolic risk factors were significantly associated with lung function impairment, who had a similar proportion of regular excise (24.6% vs. 32.6%, *p* = 0.125) and smoking (3.5% vs. 3.7%, *p* = 1.000), compared to normal lung function. These results echoed prior research findings that cardiometabolic diseases correlated with impairment in lung functioning, especially of the restrictive type [13,14]. This phenomenon might be due to the sedentary lifestyle common in many women. Previous studies indicated that adopting regular exercise had benefits for healthy lungs and in reducing cardiometabolic diseases [7,13,25]. To improve the general health status for middle-aged women in rural areas, it is strongly recommended that assessing and providing a healthy lifestyle modification, especially regular exercise as a routine assessment combined with annual health check-ups, are given importance.

Although the present study revealed that physical inactivity and lung function impairment were independently associated with metabolic syndrome, nearly half of the participants did not have healthy behaviors, such as eating vegetables and fruits, adequate daily water intake, and regular exercises. The guideline on the primary prevention of cardiovascular disease [17] recommended eating adequate vegetables and fruits and engaging in regular exercises, such as 150 min per week, for cardiometabolic disease prevention. In Spain, Gutiérrez-Carrasquilla et al. [25] found that middle-aged adults with a high adherence to the Mediterranean diet and physical activity had both higher forced vital capacity and forced expired volume in the first, as compared to those with a low adherence. Here, more than two-thirds of women adopted less exercise regularly and were found to have a higher probability of MetS. This result of middle-aged women with insufficient physical activity was similar to the same population in the Yunlin County (74%); however, it was found to be worse than the general population in Taiwan (51.8%) [15]. Clinicians and primary health-care providers can contribute to preventing the further impact of cardiometabolic diseases through early detection of unhealthy habits and cardiometabolic risk factors.

Moreover, according to the official report, and based on random sampling, Yunlin County had the highest prevalence of women with insufficient physical activity [15]. This phenomenon might explain the reason why there is a high prevalence of cardiometabolic risks exhibited in this study. Additionally, the present study revealed that 40% of women consumed water inadequately. Although adequate water intake was not shown to be significantly associated with cardiometabolic risk factors in the present finding, some work of literature indicated that because of the increased level of blood viscosity, inadequate fluid consumption, such as less than 2000 mL per day, was associated with rapid renal decline, cardiovascular disease mortality, a risk factor or predictor of cerebrovascular events, and all-cause mortality in elderly women [16,26]. Further studies should consider the importance of daily water and fluid intake and the use of more scientific measurements to estimate the dehydration condition, such as urine-specific gravity or serum osmolarity. Hence, further studies must explore the barriers of engaging in regular exercise and initiating the culture-tailored and literacy-level health-promoting programs for these underserved middle-aged women.

## 5. Limitations

There are some limitations to this study. First, the non-random sampling and limited geographical scope might limit the generalizability of the findings. Second, the prevalence of health-related behaviors, such as healthy diet, water intake, and exercise level, was determined by self-report and without the use of scientific measurement or assessment. Finally, because of a lack of history-taking on medications for hypertension or diabetes, the present findings might have underestimated the prevalence of cardiometabolic risks.

## 6. Conclusions

Regardless of the above limitations, the findings of this study revealed a high prevalence of cardiometabolic risks, metabolic syndrome, physical inactivity, and lung function impairment among rural middle-aged women. Further, lung function impairment was significantly associated with cardiometabolic risks, and physical inactivity and lung function impairment was independently associated with metabolic syndrome. It is essential to initiate a health-promoting program as a routine assessment, e.g., enhancing the benefits of exercise and reducing the barriers of adopting regular exercise in middle-aged women through empowerment strategies.

## Figures and Tables

**Table 1 ijerph-17-08067-t001:** Demographic Characteristics of the Middle-Aged Women (N = 439).

Variable	Number (%) or Mean ± SD
Demographics	
Age, years	55.9 ± 7.1
Education level (years)	8.7 ± 4.5
Working status (yes)	219 (49.9)
Live alone (yes)	28 (6.4)
Smoking (current or former)	16 (3.6)
Marital status (married)	366 (83.4)
Health-related behaviors	
Adequate vegetable ≥ 3 portions per day (≥1 and half bowel)	317 (72.2)
Adequate fruit = 2 portions per day (=1 bowel)	281 (64.0)
Adequate water ≥ 1500 cc per day	266 (60.6)
Adequate regular exercise (3 times/week; 30 min/per time)	134 (30.5)
Lung function status	
Normal	325 (74.0)
Lung function impairment ^1^	114 (26.0)
Cardiometabolic risk factor	
WC ^2^	244 (55.6)
High blood pressure ^3^	240 (54.7)
High-density lipoprotein ^4^	157 (35.8)
High fasting blood glucose ^5^	296 (67.4)
High triglyceride ^6^	143 (32.6)
Metabolic syndrome ^7^	207 (47.2)
Mean number of cardiometabolic risks	2.5 ± 1.4

^1^ FEV1, forced expiratory volume in first one second; FVC, forced vital capacity; Lung function impairment, FEV1/FVC ratio (%) < 70 or FVC (%) < 80; ^2^ WC, waist circumference > 80 cm; ^3^ Systolic/diastolic blood pressure, SBP/DBP > 130/85 mmHg; ^4^ HDL, high-density lipoprotein < 50 mg/dL (1.29 mmol/L); ^5^ fasting blood glucose > 100 mg/dL (5.6 mmol/L); ^6^ Triglyceride > 150 mg/dL (1.7 mmol/L); ^7^ Metabolic syndrome ≥ 3 risk factors; SD, standard deviation.

**Table 2 ijerph-17-08067-t002:** Cardiometabolic Risk Factors Associated with Lung Function Impairment (N = 439).

Variables	Lung Function Status		*p*
	Impairment (*n* = 114)	Normal (*n* = 325)	
Waist circumference (cm) ^1^			0.326
WC > 80	68 (59.6)	176 (54.2)	
WC ≤ 80	46 (40.4)	149 (45.8)	
Blood pressure (mmHg) ^2^			0.038
SBP/DBP > 130/85	72 (63.2)	168 (51.7)	
SBP/DBP ≤ 130/85	42 (36.8)	157 (48.3)	
High-density lipoprotein ^3^			0.013
HDL-C < 50 mg/dL (1.29 mmol/L)	52 (45.6)	105 (32.3)	
HDL-C ≥ 50 mg/dL (1.29 mmol/L)	62 (54.4)	220 (67.7)	
Fasting blood glucose ^4^			0.001
FBG > 100 mg/dL (5.6 mmol/L)	91 (79.8)	205 (63.1)	
FBG ≤ 100 mg/dL (5.6 mmol/L)	23 (20.2)	120 (36.9)	
Triglyceride ^5^			0.001
TG > 150 mg/dL (1.7 mmol/L)	52 (45.6)	91 (28.0)	
TG ≤ 150 mg/dL (1.7 mmol/L)	62 (54.4)	234 (72.0)	
Metabolic syndrome (MetS) ^6^			<0.001
MetS ≥ 3	71 (62.3)	136 (41.8)	
MetS < 3	43 (37.7)	189 (58.2)	

^1^ waist circumference (WC) > 80 cm; ^2^ Systolic/diastolic blood pressure, SBP/DBP > 130/85 mmHg; ^3^ HDL, high-density lipoprotein < 50 mg/dL (1.29 mmol/L); ^4^ Fasting blood glucose (FBG) > 100 mg/dL (5.6 mmol/L); ^5^ Triglyceride (TG) > 150 mg/dL (1.7 mmol/L); ^6^ Metabolic syndrome (MetS) ≥ 3 risk factors.

**Table 3 ijerph-17-08067-t003:** Factors Associated with Metabolic Syndrome Among Middle-Aged Women.

Predictor	Univariate Analysis		Multivariate Analysis	
	OR (95% CI)	*p*	OR (95% CI)	*p*
Age, years	1.03 (0.99–1.05)	0.065	1.02 (0.99–1.05)	0.234
Education level, years	0.97 (0.93–1.01)	0.108	1.00 (0.95–1.06)	0.864
Working status (yes)	0.77 (0.53–1.12)	0.165	0.77 (0.51–1.14)	0.192
Live alone (yes)	0.60 (0.27–1.34)	0.214	0.50 (0.20–1.25)	0.138
Marital status (married)	0.74 (0.45–1.22)	0.241	0.77 (0.43–1.38)	0.383
Vegetable ≥ 3 portions	1.07 (0.71–1.63)	0.745	1.43 (0.72–2.82)	0.307
Fruit ≥ 2 portions	0.84 (0.57–1.24)	0.370	0.73 (0.38–1.39)	0.337
Water ≥ 1500cc	0.78 (0.53–1.15)	0.209	0.95 (0.63–1.44)	0.816
Regular exercise	0.54 (0.35–0.81)	0.003	0.56 (0.36–0.87)	0.010
Lung function impairment	2.29 (1.48–3.56)	<0.001	2.12 (1.34–3.35)	0.001

OR, odds ratio; CI, confidence interval.

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
