# Peer review of "Prevalence of and Factors Associated with Cardiometabolic Risks and Lung Function Impairment among Middle-Aged Women in Rural Taiwan"

_ijerph, 2020, doi:10.3390/ijerph17218067_

Round 1

Reviewer 1 Report

The article is well written in a concise, straight-forward manner. The results are well presented with sound statistic methods. The major finding that "physical inactivity and lung function impairment were independently associated with metabolic syndrome "supports the statement why Exercise Matters for Middle-aged Women".  However, it's intuitive to wonder if there is any association between physical inactivity and lung function since both factors are independently associated with the metabolic disorder. It might further complicate the questions, therefore the findings. But if possible to know, it would consolidate the statement why exercise matters (most).   

One specific question:

Line 102. “...invasive spirometer…”. Please specify the manufacture model of the “invasive” spirometer.

Reviewer 2 Report

This work assessed the prevalence of lung function and cardiometabolic risks in a transversal sample of middle-aged women from rural areas of Taiwan. The investigated the factors associated with both impaired lung function and metabolic syndrome. The work is original and relevant, and the manuscript was well-written. However, there are several issues that must be addressed, as described below.

  1. The title of the study is not adequate because the work was not aimed for early detection of lung function impairment (i.e., it is not a diagnostic study). The title should reflect clearly the scope of the study, for instance: "Prevalence and risk factors associated with cardiometabolic risks and lung function impairment and associated risk factors in rural Taiwan middle-aged women". It is also not appropriate the first part of the current title: “Why exercise matters for middle-aged women” since the study design cannot inform of causality.
  2. The aim of the study should be mentioned explicitly at the end of the Introduction section.
  3. Section 2.3, Measurements, lines 98 to 100, the question regarding health-related behaviors. Was this only one question regarding four different health-related behaviors? That would entail some difficulties in the analysis. Please describe it in more detail.
  4. Section 2.4 Data analysis.
    • Please describe how the continuous variables were tested for normal distribution, and which tests were used to compare between groups.
    • Regarding the logistic regression analysis, the authors mention a multinomial logistic regression of lung function impairment, but the results show comparisons only between two groups of lung function impairment: normal or impairment. Did you perform a binary logistic regression instead of a multinomial logistic regression? Please clarify.
  5. Table 3. Change “Univairate” with “Univariate”
  6. Conclusions section. The conclusions section should be based mostly on the actual findings of the present work. All the comments starting in the second sentence (line 210) are clinical implications or recommendations, cannot be considered conclusions of the present study, and should be addressed in the Discussion section.

Reviewer 3 Report

This paper identified potential to improve healthy life expectancy of a rural subpopulation of middle-aged women in in Taiwan, which appears to be less than the national healthy life expectancy.  The paper identified modifiable lifestyle factors that could be targeted to reduce prevalence and severity of metabolic syndrome associated cardiovascular disease risk and declining pulmonary function.

SI units for blood chemistry markers should be used in text and captions.  US conventional (mg/dL) units can accompany SI lab values.

I've subsequently noticed that the links under references #4 and #15 are duplicated and incorrect.

It seems odd that the authors captured pulmonary function (FEV1/FVC) but did not capture smoking status.

Recent studies have found a relationship between smoking and beta cell failure that seems to define a non-insulin-resistant subtype of metabolic syndrome.

The article will appear remiss if published without reanalysis that considers smoking status of women that were included in the Yunlin County vs general Taiwan populations.

Round 2

Reviewer 2 Report

All the concerns have been addressed properly in the revised manuscript.

There are no further comments.